# Antidepressant prescriptions, discontinuation, depression and perinatal outcomes, including breastfeeding: A population cohort analysis

**Sue Jordan**[1☯]*, **Gareth I. Davies**[2☯], **Daniel S. Thayer**[2☯], **David Tucker**[1,3☯],
**Ioan Humphreys**[1☯]

**1** College of Human and Health Sciences, Swansea University, Swansea, United Kingdom, **2** School of Medicine, Swansea University, Swansea, United Kingdom, **3** Public Health Wales, Cardiff, United Kingdom

☯ These authors contributed equally to this work.
* s.e.jordan@swansea.ac.uk

**Data Availability Statement:** All relevant data are within the manuscript and its Supporting Information files. Additional data are available in the EUROmediCAT report and its appendices

## Abstract

### Objectives

To explore associations between exposure to antidepressants, their discontinuation, depression [medicated or unmediated] and preterm birth [<37 and <32 weeks], small for gestational age (SGA) [<10th and <3rd centiles], breastfeeding [any] at 6–8 weeks.

### Methods

**Design:** A population-based cohort study.

**Setting:** The Secure Anonymised Information Linkage [SAIL] databank in Wales, linking maternal primary care data with infant outcomes.

**Participants:** 107,573, 105,331, and 38,725 infants born 2000–2010 with information on prematurity, SGA and breastfeeding respectively, after exclusions.

**Exposures:** Maternal antidepressant prescriptions in trimesters 2 or 3, discontinuation after trimester 1, recorded diagnosis of depression [medicated or unmediated] in pregnancy.

**Methods:** Odds ratios for adverse pregnancy outcomes were calculated, adjusted for smoking, parity, socio-economic status, and depression.

### Results

Exclusive formula feeding at 6–8 weeks was associated with prescriptions in trimesters 2 or 3 for any antidepressants (adjusted odds ratio [aOR] 0.81, 95% confidence intervals 0.67–0.98), SSRIs [aOR 0.77, 0.62–0.95], particularly higher doses [aOR 0.45, 0.23–0.86], discontinuation of antidepressants or SSRIs after trimester 1 (aOR 0.70, 0.57–0.83 and 0.66, 0.51–0.87), diagnosis of depression aOR 0.76 [0.70–0.82], particularly if medicated (aOR 0.70, 0.58–0.85), rather than unmedicated (aOR 0.87, 0.82–0.92). Preterm birth at <37 and <32 weeks' gestation was associated with diagnosis of depression (aOR 1.27, 1.17–1.38, and 1.33, 1.09–1.62), particularly if medicated (aOR 1.56, 1.23–1.96, and 1.63, 0.94–2.84); birth at <37 weeks was associated with antidepressants, (aOR 1.24, 1.04–1.49). SGA <3rd

(reference 33 - publicly available). The datasets generated and analysed during the current study are not publicly available, because the anonymised data, held by SAIL, can only be accessed within the SAIL secure remote access environment within the context of an approved project, following governance review. No patient level data are available under the terms of ethical and governance reviews. No participant consent is obtained for population level studies. Individual records for all databases were anonymised, and individual patient data cannot be publicly deposited or fully shared upon request. They are only available directly from the database providers. All interested parties are able to apply to obtain the data in the same way as the current investigators. Data used in the study can be accessed within the SAIL secure environment subject to approval by the SAIL Information Governance Review Panel. The URL for the SAIL databank is: https://saildatabank.com/, and the non-author contacts are Cynthia McNerney, Information Governance Manager, The SAIL Team, Swansea University: c.l. mcnerney@swansea.ac.uk and Charlotte Arkley, Administrator, SAIL Databank c.r.arkley@swansea. ac.uk. Dataset information (names of datasets and provenance) is provided under 'Methods, subsection Setting'. As indicated in the text, further details are in reference 33. The corresponding author will endeavour to meet requests for further data and information. We confirm that data that are not publicly available are not part of the minimal data set.

**Funding:** This paper was developed from the EUROmediCAT project, and uses the cohort identified in that project. The analyses presented here were completed outside the funded period. Financial support for the EUROmediCAT study was provided by the European Union under the 7th Framework Program [grant agreement HEALTH-F5-2011-260598]. Start date: 1 March 2011. Duration: 48 months. Coordinator Prof. Helen Dolk, University of Ulster. Further information can be found at www.euromedicat.eu The paper is based on data in the all-Wales SAIL databank, which is supported by UK Research and Innovation funding to Swansea University through an Administrative Data Research Centre grant (2018-2921), project reference: ES/S007393/1, Principal Investigator: Professor David Ford.

**Competing interests:** The authors have declared that no competing interests exist.

centile was associated with antidepressants (aOR 1.43, 1.07–1.90), and SSRIs (aOR 1.46, 1.06–2.00], particularly higher doses [aOR 2.10, 1.32–3.34]. All adverse outcomes were associated with socio-economic status and smoking.

## Implications

Exposure to antidepressants or depression increased risks of exclusive formula feeding at 6–8 weeks, and prescription of antidepressants was associated with SGA <3rd centile. Prescription of antidepressants offers a **useful marker** to target additional support and additional care before and during pregnancy and lactation.

## Introduction

Some 20% women aged 16–54 report at least one common mental health problem: this figure [point prevalence of 19.5–25.2%] has remained unchanged since 2000. [1] The prevalence of depression during pregnancy is reported as between 6 and 13%,[2, 3] and around 10% pregnant women develop a depressive illness during pregnancy or *postpartum*, with a further 16% developing a self-limiting depressive reaction. [4]

Selective serotonin reuptake inhibitors [SSRIs] are the most commonly prescribed antidepressants: [5,4] 2.8% [6] to 10.2%[7] pregnant women are prescribed SSRIs during pregnancy, with marked variation across Europe. [8] SSRIs, and their metabolites, cross the placenta, [9] and appear in cord blood [10, 11] in proportion to dose administered; [12] foetal exposure is prolonged by accumulation in amniotic fluid. SSRIs, and some other antidepressants, act on the crucial serotonin transporter [SERT, aka 5HTT, SLC6A4, OMIM 182138], increasing the bioavailability of serotonin [5HT] in many tissues, including the placenta. [13] Serotonin-induced vasoconstriction [11, 14, 15] reduces placental blood flow, leaving the foetus vulnerable to intra-uterine growth retardation. Serotonin may also promote preterm birth by its uterotonic actions. [16]

The dose-response relationship between SSRI prescriptions in trimester 1 and the adverse outcome 'major congenital anomaly and/ or stillbirth' [17] is a concern, but the effects of SSRI exposure, discontinuation and depression on the range of pregnancy outcomes, including lactation, need to be considered together. Observation studies indicate increased risks of spontaneous abortion [18] low birth weight [19], prematurity [20, 21, 22], admission to neonatal special care facilities, [23] gestational hypertension, [24] and persistent pulmonary hypertension in neonates. [25, 26] However, despite biological plausibility, not all studies report an increase in preterm birth, [27] and poor parental perinatal mental health can adversely affect childhood outcomes. [27]

Many studies are unable to account for the effects of underlying mental illness, discontinuation of medication, social stress or smoking and some associations with adverse outcomes may be shared with non-SSRI antidepressants [28] or depressive illness. [29] Consequently, the key question for women and clinicians—the harm to benefit balance of starting, stopping or continuing antidepressant pharmacotherapy [5]—remains unanswered. Accordingly, the aim of this study is to investigate any associations between antidepressants, SSRIs at standard and higher doses, their discontinuation after trimester 1, and depression [medicated or unmedicated] and the range of adverse outcomes important to women: preterm birth, intra-uterine growth as SGA and, for the first time, breastfeeding [any] at 6–8 weeks, complementing reports of increased prevalence of congenital anomalies [17].

## Methods

A population-based cohort was built from prospectively collected routine NHS data and analysed retrospectively.

### Ethics

The Secure Anonymised Information Linkage [SAIL] Databank Information Governance Review Panel [IGRP] approved the study on behalf of the National Research Ethics Service, Wales on 24th March 2011. Data were irrevocably anonymised and obtained with permission of the relevant Caldicott Guardian and Data Protection Officer [30].

### Setting

Data were extracted from existing routinely collected data in SAIL, housed in Swansea University. Within SAIL, we linked primary care records, including prescriptions, for the ~40% of the population whose general practitioners (GPs) had agreed to share data with SAIL, without payment, by 2014. [31,32] to: the Office of National Statistics births and deaths register, the National Community Child Health Database [NCCHD] [http://www.wales.nhs.uk/news/28291], the Patient Episode Database for Wales [http://www.wales.nhs.uk/document/176173], CARIS [Congenital Anomaly Register and Information Service for Wales] [http://www.caris.wales.nhs.uk/home]. Databases were linked by a trusted third party [NHS Wales Informatics Service [NWIS], [http://www.wales.nhs.uk/sitesplus/956/home], using unique personal identifiers, which remained undisclosed to researchers, ensuring anonymity. [33]

### Population

The study population included all births in Wales after 24 gestational weeks between 1st January 2000 and 31st December 2010, with linked maternal prescription data. Infants were included where the associated maternal ID could be linked with the primary care dataset, which was dependent on the general practice, and the record was complete. [33] We included all infants where the woman was present in the linked database with primary care prescription information 91 days before last menstrual period [LMP] to birth. Information on start of pregnancy was obtained from ultrasound scan data recorded in the NCCHD. [33]

### Exposure

Exposure was defined as one or more prescription for an antidepressant issued between 92 days after the first day of LMP and birth [Table 1]. We based our timeframe on prescription duration [typically 90 days] and relevant pharmacokinetic parameters: for example, elimination of the active metabolite of fluoxetine can take ~40 days in adults, [34] and longer in the embryo or foetus. [35] Antidepressants were investigated according to anatomical, therapeutic, chemical [ATC] classification grouped: a] all SSRIs [N06AB]; b] all serotonin and norepinephrine reuptake inhibitors (SNRIs) [N06AX]; c] all antidepressants, including SSRIs and SNRIs [N06A].

Depression was defined as any diagnosis of depression in the woman's record before the end of trimester 1 [91 days after 1st day of LMP], recorded by the GP using Read codes, version 2 [17,36, 37] (S1 File). Vulnerability and kindling hypotheses [38] suggest that any episode of depression may predispose to recurrence during pregnancy [39] with associated stressor-induced release of inflammatory cytokines, and any episode of depression permanently changes the hippocampus, prefrontal cortex neurochemistry and fronto-cingulate connections: [38] accordingly, we did not time-limit depression exposure. Medicated and unmedicated depression were defined as a recorded diagnosis of depression, with / without any

**Table 1. Classification of exposures.**

| Exposure | ATC | Timeframe |
|---|---|---|
| SSRI [any]<br>≥1 SSRI [any] prescription in trimesters 2 or 3 | N06AB | From 92 days after the first day of LMP to birth |
| High dose SSRI<br>≥1 high dose SSRI [any] prescription t2-3 [defined as tablet sizes: 60mg fluoxetine, 40mg citalopram, 30mg paroxetine, 100mg sertraline, 20mg escitalopram] | N06AB | From 92 days after the first day of LMP to birth |
| Antidepressant [any]<br>≥1 Antidepressant prescription trimesters 2 or 3 t2-3* | N06A | From 92 days after the first day of LMP to birth |
| SNRI [any]<br>≥1 prescription for an SNRI in trimesters 2 or 3 | Venlafaxine N06AX16, reboxetine N06AX18, duloxetine N06AX21 | From 92 days after the first day of LMP to birth |
| Depression diagnosis in the primary care record whilst on the database | - | Any time before the end of trimester 1 [91 days after 1st day of LMP] |
| Unmedicated depression | | Depression diagnosis recorded [as above] but no antidepressant prescribed in trimesters 2 or 3 [t2-3]. |
| Medicated depression | N06A | Depression diagnosis recorded [as above] plus an antidepressant in trimesters 2 or 3 [t2-3]. |
| Discontinued SSRI:<br>≥1 SSRI prescription in trimester 1, but not in trimesters 2 or 3 | N06AB | Prescription between 1st day LMP and 91 days after 1st day LMP] and no further prescriptions recorded throughout pregnancy |
| Discontinued antidepressant:<br>≥1 prescription of any antidepressant in trimester 1, but not in trimesters 2 or 3 | N06A | Prescription between 1st day LMP and 91 days after 1st day LMP] and no further prescriptions recorded throughout pregnancy |

LMP—last menstrual period, SSRI–selective serotonin reuptake inhibitor, SNRI–serotonin and noradrenaline reuptake inhibitor

Medicines were defined by ATC codes, and then matched to version 2 Read codes in the GP database, using reference data provided by NHS Digital Technology Reference Data Update Distribution [https://isd.digital.nhs.uk/trud3/user/guest/group/0/home].

Depression was captured from Read codes https://www.datadictionary.nhs.uk/web_site_content/supporting_information/clinical_coding/read_coded_clinical_terms.asp?shownav=1 listed in S1 File.

antidepressant prescribed in trimesters 2 or 3. Discontinuation was defined as ≥1 prescription during trimester 1 and no further prescriptions recorded throughout pregnancy [Table 1]. Other indications for antidepressants recorded were: post-traumatic stress disorder [PTSD], obsessive compulsive disorder [OCD], panic disorder, bulimia, general and social anxiety disorders. We ascertained whether women had been admitted to hospital with a mental health diagnosis or had contacted community mental health teams [CMHTs] before [any time], during or up to 1 year after pregnancy.

**Dose** was not directly available, so tablet and capsule strengths were taken as proxies. We classified high dose SSRI exposure as prescription of: 60mg fluoxetine, 40mg citalopram, 30mg paroxetine, 100mg sertraline, 20mg escitalopram, based on tablet/ capsule sizes quoted in the British National Formulary. [40] Smaller tablets and capsules were classified as 'other dose' [low or medium]. [33]

## Outcomes

Prematurity was defined as <37, and extreme prematurity as <32 completed weeks' gestation. [41] Growth centiles were calculated from WHO standards for the UK, and infants below the 10th and 3rd centiles were identified; the latter category is defined as 2 standard deviations below the median. [42]

Breastfeeding [any, even if supplemented by formula feeding] at birth and 6–8 weeks is routinely recorded by health visitors, and this information is transferred to NCCHD; data

collection is more complete in some Health Boards than others. [43] Breastfeeding at birth is considered an indirect measure, which may represent intention rather than practice, and may not indicate successful breastfeeding; [44] we report data at 6–8 weeks. Data were available from 2004, reducing the sample size.

## Confounding

To minimise **confounding by co-exposure**, we excluded pregnancies known to be at increased risk of adverse outcomes. We achieved a relatively homogeneous population by excluding from the main analysis infants: 1) with major congenital anomalies [as defined by EUROCAT; [45] 2) from multiple pregnancies; 3) stillborn; 4) whose mothers were prescribed medicines more closely associated with adverse outcomes than antidepressants in trimester 1 or the quarter preceding pregnancy: anti-epileptic drugs [AEDs] [N03]; [46] coumarins [B01AA], mainly warfarin [47]; insulins [A10A]; [48] and 5) whose mothers had any record of: heavy alcohol use and/or substance misuse. [49] We did not exclude moderate alcohol use as this is not known to affect perinatal outcomes, [49] and may be inconsistently recorded. To minimize **confounding by indication,** we adjusted analyses for depression [any recorded diagnosis], [38] and investigated depression, both medicated and unmedicated, discontinuation of medication and higher doses separately. We also adjusted for socioeconomic status [SES], as Townsend fifths (Tables A1, A2 in S2 File), parity, smoking [as 'yes' or 'no'], and year of birth. In this cohort, SES is associated with antidepressant prescriptions, depression and maternal age at birth. [17,50] Where confounders are correlated, the odds ratios are less vulnerable to bias [51]. Therefore, in view of the low numbers in some outcomes, to reduce co-linearity with SES and parity, age [a non-modifiable risk factor] was not entered into the regression analyses. There were too few women aged >39 at birth to explore this potentially high risk group separately.

## Statistical analysis

We explored associations between prematurity, SGA and breastfeeding and: a) ≥1 prescription in trimesters 2 or 3 of any antidepressant, and SSRIs at any and high doses; b) discontinuation of SSRI or antidepressant after trimester 1; c) depression, medicated or unmedicated. Outcomes with sufficient numbers of exposed pregnancies were explored by multivariate logistic regression, backwards likelihood ratio [52], with covariates SES [53], parity, smoking, year of birth, and, where appropriate, depression using SPSS version 25 for windows [IBM Corp 2011]. [54] Interaction variables 'depression*SSRI' or 'depression*antidepressant' were entered into the models to test the independence of the diagnosis and the prescription. Where associations were statistically significant, numbers needed to harm [NNH] were calculated, to aid interpretation of findings. We explored demographic differences between the women with and without breastfeeding data at 6–8 weeks.

## Results

After exclusions, 107,573, 105,331, and 38,725 infants available for analysis on prematurity, SGA and breastfeeding [Fig 1]. Adverse outcomes were more prevalent and breastfeeding rates were lower in infants exposed to insulin, AEDs, coumarins substance misuse or heavy alcohol use [Tables B1, B2 in S2 File]. Some 2% women prescribed antidepressants were co-exposed to AEDs, and exposure to insulin, smoking, substance misuse or heavy alcohol use was more prevalent than in the population. Co-exposures with diagnosed depression was slightly lower. Both depression and antidepressant prescriptions were more prevalent amongst the most economically deprived and the most obese [BMI>30] [Tables A1, A2 in S2 File].

In trimesters 2 or 3, 1625 [1.5%] women were prescribed SSRIs, [538 at high doses], and 2043 [1.9%] an antidepressant. Of these, 836 [51.4%] and 1048 [51.3%] had a diagnosis of depression. 12,748 women had been diagnosed with depression at some point. Many women diagnosed with post-traumatic stress disorder [PTSD], obsessive compulsive disorder [OCD], panic disorder, bulimia, general and social anxiety disorders were also diagnosed with depression. Of the 2394 women diagnosed with any of these conditions, but not depression, 141 [5.89%] were prescribed an antidepressant in trimesters 2 or 3, indicating that 854 [2043-1048-141] [41.80%] recipients of antidepressants had no recorded indication for their prescriptions.

Most (4252/6295, 67.55%) women prescribed antidepressants in trimester 1 had no further prescriptions in trimesters 2 or 3: of these, 2076 [48.8%] had a diagnosis of depression, and 2285 were prescribed an SSRI. In the entire cohort (107,573), nine women were admitted to hospital with a mental health diagnosis during pregnancy, and 18 in the year following childbirth. Eleven or the 27 women admitted had discontinued an antidepressant after trimester 1, representing 0.26% of the 4252 discontinuing. Some [numbers too low for disclosure] had contacted their CMHTs. Of the 4252 women discontinuing antidepressants, 163 (3.93%) had contacted CMHTs during, and 127 (2.99%) after pregnancy.

All exposures were associated with increased prevalence of at least some adverse outcomes. Not all associations were statistically significant. There were low numbers of infants in the extreme prematurity, <3rd centile, high dose and medicated depression categories. Economic deprivation, smoking and primiparity were associated with SGA and prematurity; only breastfeeding was associated with year of birth [Tables C-E in S2 File].

**Preterm birth,** before 37 and 32 weeks' gestation, was associated with a diagnosis of depression [aOR 1.27, 1.17–1.38, NNH 69, 52–101 and 1.33, 1.09–1.62, NNH 389, 222–1451], particularly medicated depression [aOR 1.56, 1.23–1.96], NNH 30, 20–62, and aOR 1.63, 0.94–2.84]. Birth at <37 weeks' was associated with any antidepressant in trimesters 2 or 3 [1.24, 1.04–1.49, NNH 44, 29–90]. Associations with SSRI prescription were not statistically significant. Prevalence of birth at <32 weeks' was highest with medicated depression and high dose SSRIs, but associations did not reach statistical significance. There was no increased risk if antidepressants were discontinued after trimester 1 [Tables 2 and 3].

**Birth weight below 3rd centile** was associated with SSRI or any antidepressant prescription in trimesters 2 or 3 [aOR 1.46, 1.06–2.00, NNH 81, 48–261 and 1.43, 1.07–1.90, NNH 90, 54–275], particularly high dose SSRIs [aOR 2.10, 1.32–3.34, NNH 40, 24–130]. Associations with diagnosed depression or discontinuation of medication after trimester 1 were statistically insignificant. SGA <10th centile was only statistically significantly associated with discontinuation of antidepressants [Tables 4 and 5].

**At 6–8 weeks, breastfeeding** [any] was less prevalent amongst those prescribed SSRIs [aOR 0.77, 0.62–0.95, NNH 9, 7–12], particularly high doses [aOR 0.45, 0.23–0.86, NNH 9, 6–18], or any antidepressants in trimesters 2 and 3 [aOR 0.81, 0.67–0.98, NNH 10, 8–13]. Depression was an equally important predictor of exclusive formula feeding [aOR 0.76, 0.70–0.82, NNH 12, 11–14], and discontinuation of SSRIs or antidepressants after trimester 1 offered no benefit in terms of predicting breastfeeding success [aOR 0.66, 0.51–0.87, NNH 11, 9–16 and 0.70, 0.57–0.85, NNH 10, 8–13]. Medicated depression appeared a stronger predictor of exclusive formula feeding than unmedicated depression [Table 6]. High SES, non-smoking and nulliparity predicted breastfeeding at 6–8 weeks. Associations with breastfeeding at birth were similar [Tables D, E in S2 File].

**Interaction terms** between depression and prescribed medicines were statistically significant for: discontinuing SSRIs and breastfeeding, gestation <37 weeks, and <10th centile; high dose SSRIs and breastfeeding, <37 weeks; discontinuing any antidepressant and breastfeeding, <37 weeks and birth weight <10th centile. Interaction terms were not statistically significant for

**Study Flow CONSORT Diagram**

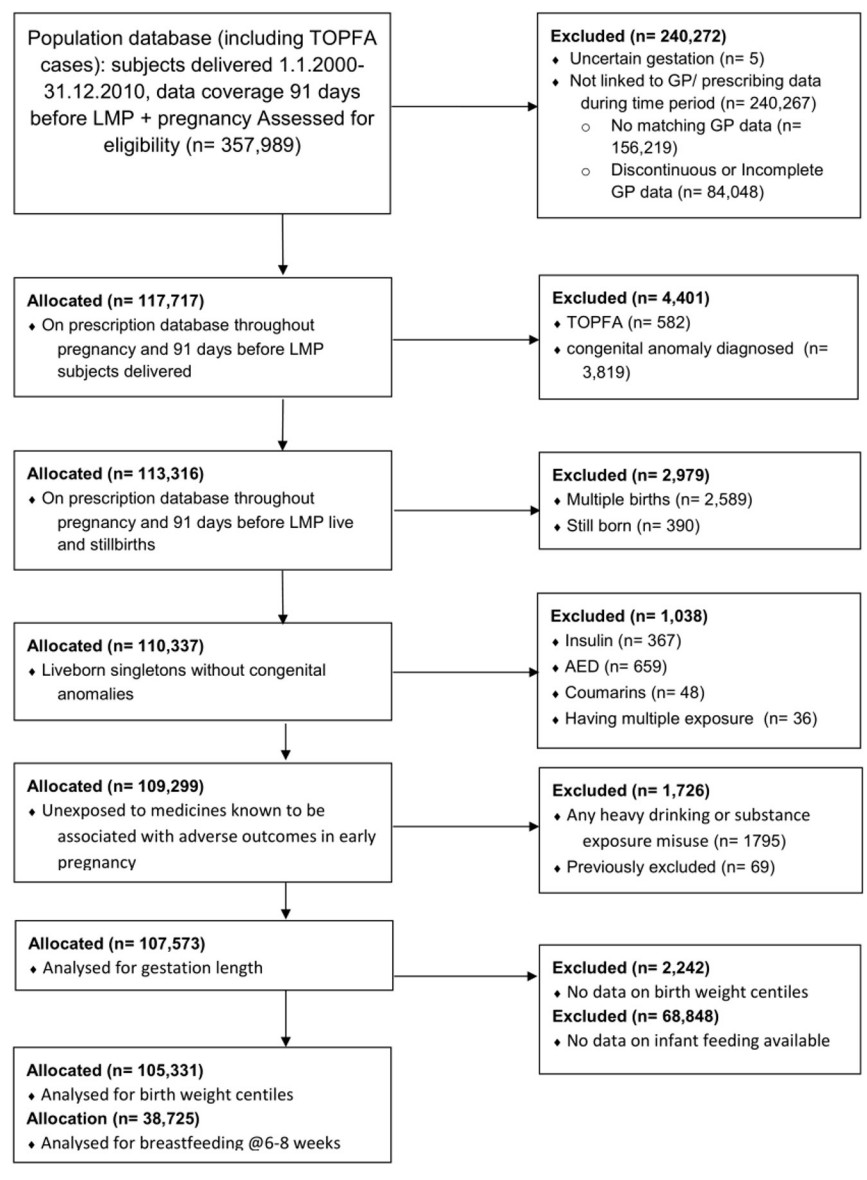

**Fig 1. Study flow diagram.**

prescription of SSRIs or antidepressants and prematurity, SGA or breastfeeding, indicating that the impact of depression and the prescriptions were independent of each other (Tables 2–6).

Seventy-three infants were exposed to **SNRIs** in trimesters 2 & 3, including 49 whose mothers had been diagnosed with depression. None of these was born before 32 weeks, <5 were below the 3rd centile, OR >2, but not statistically significant, and 6 were breastfeeding at 6–8 weeks, OR 2.39 [0.99–5.75]. Unadjusted analysis indicated a significant association with birth at <37 weeks, OR 2.93 [1.54–5.56], and a non-significant association with <10th centile OR 1.30 [0.52–3.21]. [Numbers were too low for adjusted analyses.] Too few women were diagnosed with PTSD, OCD, panic disorder, bulimia, general and social anxiety disorders for exploration. Women with breastfeeding data were older than those without [mean ages 28.48

**Table 2. Preterm birth @<37 weeks' gestation (n = 107573).**

| Exposures | Exposed n [%] | Unexposed n [%] | Unadjusted OR [95% CI] | Adjusted* OR [95% CI] | NNH | Interaction term with depression variable |
|---|---|---|---|---|---|---|
| SSRI [any N06AB] in trimesters 2 or 3 | 124/1625 [7.63] | 6033/105,948 [5.69] | 1.37[1.14–1.65] | 1.19[0.97–1.46] | Ns | 1.03[0.68–1.54] |
| SSRI high dose in trimesters 2 or 3 | 38/538 [7.06] | 6119/107,035 [5.72] | 1.25[0.90–1.75] | 1.05[0.72–1.53] | Ns | 1.27[1.17–1.37] |
| Antidepressant [any N06A] in trimesters 2 or 3 | 163/2043 [7.98] | 5994/105,530 [5.68] | 1.44[1.24–1.63] | 1.24 [1.04–1.49] | 44 [29–90] | 0.90[0.63–1.30] |
| SSRI in trimester 1 but not in trimesters 2 or 3 | 165/2285 [7.22] | 5992/ 205,288 [5.69] | 1.29[1.10–1.51] | 1.00[0.77–1.30] | Ns | 1.28[1.18–1.38] |
| Antidepressant in trimester 1 but not in trimesters 2 or 3 | 306/4252 [7.2] | 5851/103,321 [5.66] | 1.29[1.15–1.46] | 0.99[0.82–1.21] | Ns | 1.28[1.19–1.39] |
| Depression unmedicated | 812/11,700 [6.94] | 5345/95,873 [5.58] | 1.26[1.17–1.36] | 1.23[1.13–1.34] | 74 [55–115] | NA |
| Depression medicated | 95/1048 [9.06] | 6062/10,652 [5.69] | 1.65[1.34–2.04] | 1.56[1.23–1.96] | 30 [20–62] | NA |
| Depression diagnosed | 907/12,748 [7.11] | 5250/94,034 [5.54] | 1.30[1.22–1.41] | 1.27[1.17–1.38] | 69 [52–101] | NA |

*Adjusted for parity, smoking, socio-economic status [SES] as Townsend fifth [quintile].

Exclusions from analysis: all congenital anomalies, terminations of pregnancy for foetal anomalies [TOPFA], stillbirths, multiple births [twins, triplets and quadruplets [no higher multiples in the dataset]], ≥1 prescription for insulin, anti-epileptic drugs [AEDs] or coumarins in the quarter preceding pregnancy and trimester 1, heavy drinking/substance misuse [any record].

Deprivation [Townsend] scores, ranks and fifths are based on geographical area of residence, using Lower Super Output Areas [LSOAs] defined by residential postcodes. This measure of material deprivation is calculated from rates of unemployment, vehicle ownership, home ownership, and overcrowding]. [54]

Abbreviations and definitions are listed in Table 1.

OR odds ratio, CI confidence intervals, NNH number needed to harm, Ns not statistically significant.

**Table 3. Preterm birth @<32 weeks' gestation (n = 107573).**

| Prescriptions | Exposed n [%] | Unexposed n [%] | Unadjusted OR [95% CI] | Adjusted* OR [95% CI] | Interaction term with depression variable |
|---|---|---|---|---|---|
| SSRI [any N06AB] in trimesters 2 or 3 | 20/1625 [1.23] | 913/105,948 [0.86] | 1.43 [0.92–2.24] | 1.24 [0.76–2.03] | 1.04(0.39–2.81) |
| SSRI high dose in trimesters 2 or 3 | 7/538 [1.3] | 926/107035 [0.87] | 1.51 [0.72–3.19] | 1.74 [0.55–5.46] | 0.69[0.17–2.82] |
| Antidepressant [any N06A] in trimesters 2 or 3 | 23/2043 [1.13] | 910/105,530 [0.86] | 1.31 [0.86–1.99] | 1.15 [0.73–1.81] | 1.31(0.73–2.34) |
| SSRI in trimester 1 but not in trimesters 2 or 3 | 25/2285 [1.09] | 908/105288 [0.86] | 1.27 [0.85–1.90] | 0.88 [0.48–1.60] | 1.09[0.54–2.20] |
| Antidepressant in trimester 1 but not in trimesters 2 or 3 | 48/4252 [1.13] | 885/103,321 [0.86] | 1.32 [0.99–1.77] | 1.11 [0.80–1.55] | 0.85[0.44–1.65] |
| Depression unmedicated | 127/11,700 [1.09] | 806 [0.84] | 1.29 [1.07–1.56] | 1.29[1.05–1.58] | na |
| Depression medicated | 15/1048 [1.43] | 918/10,652 [0.86] | 1.67[1.00–2.79] | 1.63 [0.94–2.84] | na |
| Depression diagnosed | 139–142**/12,748 [1.09–1.11] | 791/94,034 [0.84] | OR> 1 statistically significant | 1.33 [1.09–1.62] | na |

Exclusions, definitions and abbreviations as in Table 2

*Adjusted for parity, smoking, socio-economic status [SES] as Townsend fifth [quintile].

NNH for depression diagnosed and unmedicated 409 [227–2070].

** Numbers have been blurred to avoid revealing a number <5 in another cell.

**Table 4. SGA, birth weight <10th centile (n = 105331).**

| Exposures | Exposed n [%] | Unexposed [n %] | Unadjusted OR [95% CI | Adjusted* OR [95% CI] | Interaction term with depression variable |
|---|---|---|---|---|---|
| SSRI [any N06AB] in trimesters 2 or 3 | 166/1587 [10.46] | 8991/103,744 [8.67] | 1.23 [1.05–1.45] | 1.09 [0.91–1.30] | 0.88(0.69–1.13) |
| SSRI high dose in trimesters 2 or 3 | 63/525 [12] | 9094/104,806 [8.68] | 1.44 [1.10–1.87] | 1.30 [0.98–1.74] | 0.85[0.47–1.53] |
| Antidepressant [any N06A] in trimesters 2 or 3 | 204/1993 [10.24] | 8953/103,338 [8.66] | 1.20 [1.04–1.39] | 1.07 [0.91–1.25] | 0.96(0.70–1.33) |
| SSRI in trimester 1 but not in trimesters 2 or 3 | 229/2234 [10.25] | 8928/103,097 [8.66] | 1.21 [1.05–1.38] | 1.27(1.03–1.57) | 0.76(0.56–1.03) |
| Antidepressant in trimester 1 but not in trimesters 2 or 3 | 418/4156 [10.06] | 8739/101,175 [8.64] | 1.18 [1.07–1.31] | 1.23 [1.04–1.44] | 0.79[0.62–0.99] |
| Depression unmedicated | 1070/11,435 [9.36] | 8087/93,896 [8.61] | 1.10 [1.02–1.17] | 1.04[0.97–1.12] | Na |
| Depression medicated | 107/1024 [10.45] | 9050/104,307 [8.68] | 1.23 [1.00–1.50] | 1.13 [0.91–1.41] | Na |
| Depression diagnosed | 1177/12,459 [9.45] | 7980/92,872 [8.59] | 1.11 [1.04–1.18] | 1.05 [0.98–1.13] | Na |

Exclusions, definitions and abbreviations as in Table 2

*Adjusted for parity, smoking, socio-economic status [SES] as Townsend fifth [quintile].

NNH for any antidepressant discontinuation 71 [43–204].

[6.09] vs. 28.04 [6.04] years, mean difference 0.44 [0.36–0.51], t 11.36, df 79659, p<0.001], more deprived [mean Townsend score 0.40 [3.24] vs. 0.21 [3.12], mean difference 0.20 [0.16–0.24], t 9.65, df 77578, p<0.001], and less likely to be primiparous [16095/38725, 41.6% vs. 29388/68848, 42.7% OR 0.96 [0.93–0.98].

**Table 5. SGA, birth weight <3rd centile (n = 105331).**

| Exposures | Exposed n [%] | Unexposed [n %] | Unadjusted OR [95% CI] | Adjusted* OR [95% CI] | Number Needed to Harm | Interaction term with depression variable |
|---|---|---|---|---|---|---|
| SSRI [any N06AB] in trimesters 2 or 3 | 49/1587 [3.09] | 1918/103,744 [1.85] | 1.69 [1.27–2.26] | 1.46 [1.06–2.00] | 81 [48–261] | 1.11(0.59–2.11) |
| SSRI high dose in trimesters 2 or 3 | 23/525 [4.38] | 1944/104,806 [1.85] | 2.42 [1.59–3.69] | 2.10 [1.32–3.34] | 40 [24–130] | 0.76[0.29–2.01] |
| Antidepressant [any N06A] in trimesters 2 or 3 | 59/1993 [2.96] | 1908/103,338 [1.85] | 1.62 [1.25–2.11] | 1.43 [1.07–1.90] | 90 [54–274] | 1.14(0.64–2.03) |
| SSRI in trimester 1 but not in trimesters 2 or 3 | 45/2234 [2.01] | 1922/103,097 [1.86] | 1.08 [0.80–1.46] | 1.11 [0.72–1.71] | Ns | 0.69[0.36–1.33] |
| Antidepressant in trimester 1 but not in trimesters 2 or 3 | 92/4156 [2.21] | 1875/101.175 [1.85] | 1.20 [0.97–1.48] | 1.12 [0.80–1.55] | Ns | 0.90[0.65–1.24] |
| Depression unmedicated | 244/11,435 [2.13] | 1723/93,896 [1.84] | 1.17 [1.02–1.34] | 1.10 [0.95–1.27] | Ns | Na |
| Depression medicated | 30/1024 [2.93] | 1937/104,307 [1.86] | 1.59 [1.11–2.30] | 1.41 [0.95–2.09] | Ns | Na |
| Depression diagnosed | 274/12,459 [2.2] | 1639/92,872 [1.82] | 1.21 [1.06–1.38] | 1.13 [0.98–1.30] | Ns | Na |

Exclusions, definitions and abbreviations as in Table 2

*Adjusted for parity, smoking, socio-economic status [SES] as Townsend fifth [quintile].

**Table 6. Breastfeeding at 6–8 weeks (n = 38725).**

| Exposures | Exposed n [%] | Unexposed [n %] | Unadjusted OR [95% CI] | Adjusted* OR [95% CI] | NNH | Interaction term with depression variable |
|---|---|---|---|---|---|---|
| SSRI [any N06AB] in trimesters 2 or 3 | 137/645 [21.24] | 12,656/38,080 [33.24] | 0.54 [0.45–0.66] | 0.77 [0.62–0.95] | 9 [7–12] | 1.11(0.72–1.70) |
| SSRI high dose in trimesters 2 or 3 | 47/214 [21.96] | 12,746/38,511 [33.1] | 0.57 [0.41–0.79] | 0.45 [0.23–0.86] | 9 [6–18] | 2.49[11.13–5.52] |
| Antidepressant [any N06A] in trimesters 2 or 3 | 179/806 [22.21] | 12,614/37,919 [33.27] | 0.57 [0.48–0.68] | 0.81 [0.67–0.98] | 10 [8–13] | 1.25[0.86–1.83] |
| SSRI in trimester 1 but not in trimesters 2 or 3 | 207/862 [24.02] | 12,586/37,863 [33.24] | 0.64 [0.54–0.74] | 0.66 [0.51–0.87] | 11 [9–16] | 1.59[1.10–2.30] |
| Antidepressant in trimester 1 but not in trimesters 2 or 3 | 376/1619 [23.22] | 12,417/37,106 [33.46] | 0.60 [0.54–0.68] | 0.70 [0.57–0.85] | 10 [9–13] | 1.37[1.04–1.80] |
| Depression diagnosed and unmedicated in t2 and t3 | 1365/5241 [25.87] | 11,428/33,484 [34.13] | 0.68 [0.67–0.73] | 0.87 [0.82–0.92] | 13 [11–15] | Na |
| Depression medicated in t2 or t3 | 102/472 [21.62] | 12,691/38,253 [33.18] | 0.56 [0.45–0.69] | 0.70 [0.58–0.85] | 9 [7–13] | Na |
| Depression diagnosed | 1467/5713 [25.7] | 11,346/33,012 [34.31] | 0.66 [0.62–0.71] | 0.76 [0.70–0.82] | 12 [11–14] | Na |

Exclusions, definitions and abbreviations as in Table 2

*Adjusted for SES, as Townsend fifth, parity, smoking and year of birth.

## Discussion

To our knowledge, this is the first report, in the published literature, that antidepressants, particularly high dose SSRIs, depression, particularly medicated depression, and discontinuation of antidepressants adversely affect breastfeeding rates at 6–8 weeks; the effects of high doses and discontinuation were not independent of depression. Lower rates of breastfeeding at 6–8 weeks were also apparent with exposure to insulin, AEDs, coumarins, substance misuse and heavy alcohol use, multiple pregnancy and congenital anomalies (Table B in S2 File).

Intra-uterine exposure to SSRIs, antidepressants and maternal depression were associated with adverse perinatal outcomes. Birth weight <3rd centile was significantly associated with antidepressant prescriptions [particularly high dose SSRIs], but depression was not, and the association was not statistically significant if medication was discontinued after trimester 1. Gestation length was adversely affected by depression, particularly if medicated, and prescription of antidepressants. We confirmed associations between preterm birth and SGA and economic deprivation, particularly the most deprived fifth, [55] smoking, [56] and nulliparity, [57] and between breastfeeding and SES, smoking and parity, [58, 59] but aORs were <2 in most analyses (Tables C-E in S2 File).

### Preterm birth

As elsewhere, depression[20–22, 60], particularly if medicated [20–22,61] was associated with preterm birth. SNRIs were associated with gestation <37 weeks in unadjusted analyses, partially accounting for the discrepancy between any antidepressant and SSRI exposure for this outcome. Women who discontinued antidepressants after trimester 1 had no additional risk of prematurity [Tables 2 and 3]. Meta-analysis of depression and preterm birth gave a random effects OR close to ours; however, adjusted analysis was not significant. [62] Prematurity was associated with psychiatric disorders in a Scandinavian cohort [27], and, in contrast with our findings, SSRI exposure appeared protective; however, different timeframes were used for

SSRI exposure [90 days before LMP to birth], and we found no impact if prescriptions were restricted to trimester 1. Prematurity was associated with SSRI exposure in a meta-analysis of 8 studies, most of which considered trimester 1 exposure, and the aOR [1.24, 109–1.41] was similar to ours. [21] An unadjusted analysis of 13 studies gave a slightly higher OR, 1.55, 1.38–1.74. [63] Within family analysis found SSRI exposure decreased gestation by 2.3 days, 0.8–3.8., [64] which is unlikely to be clinically significant or reflected in our findings. In a meta-analysis, the effect of depression on prematurity was more marked amongst women of lower SES in the USA and developing countries, but in contrast to our findings, SES did not affect prematurity. [60]

Maternal stress, short and long-term, increases the risk of pre-term birth, possibly due to activation of the hypothalamic-pituitary-adrenal [HPA] axis and sympathomimetic responses, which are intensified in depression, [65,66] congruent with an association between depressive illness and preterm birth. The corticotrophin releasing hormone [CRH] increase observed in SSRI treated women may, in part, be due to depressive illness, [16] but higher CRH levels are reported in women prescribed SSRIs than those with unmedicated depression. [67]

### SGA

These findings support the consensus associating intrauterine growth restriction with SSRIs: [20,22,60,68] SGA <3rd centile was more prevalent amongst those exposed to antidepressants in trimesters 2 or 3, particularly at higher doses; prevalence was lower if antidepressants were discontinued after trimester 1 [92 and 45 exposed infants] [69]. Discontinuation of antidepressants was significantly associated with SGA <10th centile, but this outcome is only a modest predictor of childhood morbidity. [70] Unmedicated depression was not significantly associated with SGA, confirming meta-analyses suggesting no association between antenatal depression and SGA [66, 71] [Tables 4 and 5].

SSRI-induced vasoconstriction [11, 14, 15] may explain associations between SSRIs and low birth weight, growth restriction, [19, 22, 20, 70] persistent pulmonary hypertension, [71, 61] pregnancy-induced hypertension [72], and certain congenital anomalies and stillbirth. [17] Our findings of increased risk of SGA <3rd centile, particularly at high SSRI doses, but a statistically insignificant SSRI effect on prematurity are consistent with *in vitro* embryo studies reporting fluoxetine dose response effects on cell proliferation, migration and differentiation, but not the timing of development. [73]

### Breastfeeding

Antidepressants or SSRIs after trimester 1, discontinuation, and depression were all associated with lower breastfeeding rates at 6–8 weeks. Reduction was greater if depression was medicated or high doses of SSRIs were prescribed, but numbers were low [[102 and 47 at 6–8 weeks], and associations with high doses and discontinuation may not have been independent of depression [interaction term aOR 2.49, 1.13–5.52] [Tables 6 and E in S2 File]. Breastfeeding initiation is less likely amongst women with depression [OR 0.68, 0.61–0.76] [62], particularly if co-exposed to antidepressants in trimester 3 [aOR 0.25, 0.11–0.57] [64], [aOR 0.25, 0.11–0.56] [64], or dispensed antidepressants [aOR 0.63, 0.50–0.80] [74], but we did not locate reports of breastfeeding at 6–8 weeks.

SSRI [any dose] exposure predicted exclusive formula feeding independently of a history of depression, suggesting an underlying biological mechanism. SSRIs may delay alveolar secretary activation by 69–86 hours, due to serotonin-dependent changes in tight (inter-cellular) junctions, [75] thwarting establishment of breastfeeding. SSRI exposure in trimester 3 affects monoamine metabolism in infants, causing a dose-response increase in restlessness, tremor,

and incoordination, [10] impeding breastfeeding; this may be the mechanism underlying delays in fine motor development at 3 years [76] or autistic-like behaviours secondary to increased serotonin *post-partum*. [77] Epigenetic changes, activation of the HPA axis and transfer of cortisol and other mediators to the fetus are associated with both maternal depression and antidepressants [16,78] and their impacts on neurobehavioural development are difficult to disentangle. [77] These symptoms, and any SSRI neonatal withdrawal symptoms of irritability, may impede latching, making breastfeeding painful & difficult, promoting breastfeeding discontinuation. How insomnia caused by SSRIs [40] affects breastfeeding is unknown. Effects may be exacerbated by transfer of medication into breast milk: transfer varies with drug, dose, timing of administration and feeding and supplementary formula feeding, but is ~5–10% of adult SSRI dose. Irritability, restlessness, diarrhoea and suboptimal weight gain are reported in case series, but reports of developmental delay attributed to lactation have not been located.[79] Some women were breastfeeding at 6–8 weeks, at all doses, supporting suggestions that impact may vary with genotype. [80]

Women using prescription medicines are less likely to breastfeed, particularly if there is little information about the transfer of the medicine to breastfed infants. [81] Women prescribed SSRIs at 12 weeks' gestation are less likely to express intention to breastfeed, [82, 83] which is a powerful predictor of breastfeeding at discharge. [58] In Wales, linear relationships between economic deprivation and breastfeeding, [58, 59] and economic deprivation and SSRI prescription in pregnancy, adjusted for diagnosed depression, have been identified. [50] Given the known benefits of breastfeeding, these new data provide further evidence that prescribing patterns are contributing to the concentration of the adverse effects of medicines amongst the poorest. Our findings [NNH 10, 8–13]suggest that successfully targeting women prescribed antidepressants or with a recorded diagnosis of depression would improve breastfeeding rates by ~10%, and protect 1.3% of infants from obesity [1.3%, 0.6–1.9% population reduction] and 0.7% of women from breast cancer [0.7%, 0.3–1.1% population reduction]. [84]

## Wider context: Addressing inequalities

Preterm birth, SGA and suboptimal breastfeeding remain threats to global health. These must be minimised before UN targets to reduce neonatal mortality to <10 per 1,000 live births and the prevalence of non-communicable disease [cardiovascular disease, obesity and type 2 diabetes] can be achieved. [85] Lower family income and education intensify the impact on families of behavioural and neurodevelopmental sequelae [cerebral palsy, cognitive impairment, impaired hearing or vision], [86] and the concentration of adverse outcomes and antidepressant prescriptions amongst the most economically deprived intensifies the importance of using these findings to target support. [50, 17] Women with depression or prescribed antidepressants, insulin, AEDs or coumarins or misusing alcohol and other substances need additional breastfeeding support *post-partum*, including uninterrupted time with their infants.

The risks of harm from depression [mainly prematurity] should be considered against those of antidepressant prescription, particularly SGA <3rd centile [NNH 90, 54–274], which is associated with neonatal mortality, seizures and sepsis. [87] For women with severe depression the additional risks associated with antidepressants are outweighed by the high risk of relapse [88]. However, detailed monitoring is warranted. [89] For example, scans in trimester 3 for women prescribed antidepressants in trimesters 2 and 3 would identify growth restriction, and ensure delivery of affected infants where neonatal intensive care is available. [17] Systematic review suggests that cognitive behavioural therapy [CBT] improves depressive symptoms [90] as effectively as medication. [91] Randomization of women diagnosed with depression to SSRI rather than psychotherapy increases the risk of preterm birth, 6.8% vs 5.8%

aOR 1.17, 1.10–1.25, [21] but further exploration of the impact of talking therapies or discontinuation in early pregnancy on pregnancy outcomes is needed. [92,93]

## Limitations and strengths

We acknowledge the limitations of routine health services databases: **adherence to prescribed regimens** cannot be ascertained from prescription data [17]. However, in the absence of clinical trials, this detailed analysis of continuation and discontinuation of antidepressants and depression in pregnancy identifies a population receiving insufficient support.

**Incomplete information.**    Only data recorded by primary care professionals could be analysed. Hospital, private and internet prescribing, genetic makeup, family history, particularly paternal family history, **recreational drug use, alcohol use and substance misuse** are captured poorly in routine care, fieldwork and databases: however, the Wales database is relatively complete for the last. [17] The high proportion of missing data for BMI precluded incorporation into multivariate analyses. Ethnicity is considered too sensitive for release to researchers by Wales' trusted third parties.

**Diagnosing depression.**    Comparisons with other studies may be complicated by differences in ascertainment of diagnosis of depression, and the low numbers of hospital admissions. Depression may be under-reported in primary care records, due to inaccurate diagnosis by primary care practitioners, [94] fears of 'labeling' or stigmatizing, [72] and, possibly, incomplete record transfer from secondary care. [17] We acknowledge the risks of under-ascertainment or inconsistency between individual GPs, and the limitations of taking the absence of records as 'no problems'. The decision not to time-restrict the diagnosis of depression is based on the depression diathesis model, [36] and the reluctance of clinicians to repeatedly enter the same diagnoses. Alternative indications for anti-depressants did not account for the 42% of women, including some women prescribed higher doses, with no recorded indications for antidepressants. Manifestations of depression are not routinely and uniformly recorded in healthcare databases, and we were unable to explain why so many women discontinued their prescriptions, without needing hospitalization or CMHT contact. However, consideration of interaction terms, medicated and unmedicated depression, doses, admissions, and discontinuation makes our analyses as robust as possible.

**Potential confounding and effect mediation.**    Like all non-randomised studies, population cohort analyses are vulnerable to residual confounding [51, 95]. To address confounding by indication we analysed prescriptions and diagnoses separately and together, and checked interaction terms. Women diagnosed with depression but not receiving prescriptions may have been relatively unaffected by symptoms, and those continuing into trimesters 2 and 3 or prescribed higher doses may have experienced more severe symptoms, risking confounding by severity [95]. Exclusion of co-exposed infants from the analysis reduced confounding by co-exposure: use of insulin, AEDs, coumarins, substances and alcohol was higher amongst women with depression, medicated and unmedicated, and adverse perinatal outcomes were more prevalent following these exposures [46–49] [Tables A and B in S2 File]. We were unable to determine if any adverse outcomes were complicated by co-exposure to pre-eclampsia or pregnancy-induced hypertension, [24, 26, 96] which might have prompted preterm induction of labour or caused SGA. Congenital anomalies are associated with SGA and preterm birth [97, 98], therefore, affected infants were excluded; associations are previously reported [17].

**Multiple testing.**    We acknowledge the hazards of **multiple testing**, without correction. Our independent [predictor] variables are highly correlated. This makes standard adjustments, such as the Bonferroni method, unduly conservative, risking false negatives. [99] Our findings are consistent with biological plausibility and the literature.

**Generalization.** Data from one European country, where most of the population are in an EU convergence zone [GDP <75% of the EU mean], cannot necessarily be extrapolated to different populations, but they evidence the need for national initiatives here and in similar populations. There were too few infants at <32 weeks' gestation to draw conclusions, as elsewhere, [67] but analysis of high dose exposure assists interpretation. The restrictions of our cohort were due to incomplete GP participation and incomplete data, secondary to practices' technical failures or women moving to practices not covered by SAIL [33]. Rather than any participant self-selection bias, the breastfeeding outcome was restricted by variations in recording practices across the seven Health Boards in Wales. Had a volunteer bias been operating, we should have expected to see older, less deprived women over-represented [100]; this was not the case. Too few women were hospitalised for mental health problems to draw conclusions.

Drawing causal inferences from observation data may be imprudent. However, using these findings to target support is justified, reflecting that, for most adverse outcomes, retrospective analyses yield lower odds or risk ratios than prospective studies. [101]

## Conclusion and implications: Support before, during and after pregnancy

This analysis identified a population vulnerable to adverse outcomes, including exclusive formula feeding at 6–8 weeks. Antidepressants, particularly high dose SSRIs, increased the prevalence of SGA <3rd centile, whereas depression, particularly if medicated, affected duration of gestation. Like others [69], we found little evidence that antidepressants are protective. Discontinuing antidepressants after trimester 1 appeared to reduce risks of SGA <3rd centile, but infants remained vulnerable to exclusive formula feeding. The concentration of prescribing and adverse outcomes amongst the economically disadvantaged should encourage pre-conception monitoring *in tandem* with review of antidepressants and non-pharmacological therapies. These data indicate that prescription of antidepressants is an **important marker** for adverse outcomes, easily identified in primary care records. This could, and should, be used to trigger additional monitoring, including third trimester scans or alternative continuous monitoring technology to detect SGA. *Using prescriptions to target care before, during and after pregnancy warrants exploration as a strategy to optimize breastfeeding at 6–8 weeks, and all perinatal outcomes [17].*

## Supporting information

**S1 File. Read codes for depression.**
(DOCX)

**S2 File. Supplementary tables.**
(DOCX)

**S3 File. STROBE statement.**
(DOC)

## Acknowledgments

This study uses anonymised data held in the Secure Anonymised Information Linkage [SAIL] system, which is part of the national e-health records research infrastructure for Wales. We should like to acknowledge all the data providers who make anonymised data available for research. Data held in SAIL databases are anonymised and aggregated and have been obtained

with permission of relevant Data Protection Officers, as approved by the National Research Ethics Service, Wales. Under this agreement, disclosure of numbers <5 is not permitted. The project was approved by the SAIL Information Governance Review Panel [IGFRP] on 24th March 2011. Since the project uses only anonymised data, ethical review was deemed unnecessary.

**We should like to thank**: Hildrum Sundseth from the European Institute of Women's Health for advice as service user representative on EUROmediCAT, and Professors Helen Dolk and Joan Morris for their roles as leads of EUROmediCAT and their continuing support.

## Author Contributions

**Conceptualization:** Sue Jordan.

**Data curation:** Daniel S. Thayer.

**Formal analysis:** Sue Jordan, Gareth I. Davies, Daniel S. Thayer.

**Funding acquisition:** Sue Jordan.

**Investigation:** Sue Jordan, Gareth I. Davies, Daniel S. Thayer, David Tucker, Ioan Humphreys.

**Methodology:** Sue Jordan.

**Project administration:** Sue Jordan.

**Resources:** Sue Jordan.

**Visualization:** Sue Jordan.

**Writing – original draft:** Sue Jordan, Gareth I. Davies, Daniel S. Thayer, David Tucker, Ioan Humphreys.

**Writing – review & editing:** Sue Jordan, Gareth I. Davies, Daniel S. Thayer, David Tucker, Ioan Humphreys.

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
