## [Decision Letter · Decision Letter 0]

17 Sep 2019

PONE-D-19-17104

Antidepressant prescriptions, discontinuation, depression and perinatal outcomes, including breastfeeding: a population cohort analysis

PLOS ONE

Dear Professor Jordan,

Thank you for submitting your manuscript to PLOS ONE. After careful consideration, we feel that it has merit but does not fully meet PLOS ONE’s publication criteria as it currently stands. Therefore, we invite you to submit a revised version of the manuscript that addresses the points raised during the review process.

Two reviewers addressed several major and minor concerns about your manuscript. Please revise your manuscript carefully.

We would appreciate receiving your revised manuscript by Nov 01 2019 11:59PM. To enhance the reproducibility of your results, we recommend that if applicable you deposit your laboratory protocols in protocols.io, where a protocol can be assigned its own identifier (DOI) such that it can be cited independently in the future. For instructions see: http://journals.plos.org/plosone/s/submission-guidelines#loc-laboratory-protocols

We look forward to receiving your revised manuscript.

Kind regards,

Kenji Hashimoto, PhD

Academic Editor

PLOS ONE

Journal Requirements:

Reviewers' comments:

Reviewer's Responses to Questions

**Comments to the Author**

1. Is the manuscript technically sound, and do the data support the conclusions?

Reviewer #1: Yes

Reviewer #2: Yes

2. Has the statistical analysis been performed appropriately and rigorously? 

Reviewer #1: I Don't Know

Reviewer #2: I Don't Know

3. Have the authors made all data underlying the findings in their manuscript fully available?

Reviewer #1: Yes

Reviewer #2: Yes

4. Is the manuscript presented in an intelligible fashion and written in standard English?

Reviewer #1: Yes

Reviewer #2: Yes

5. Review Comments to the Author

Reviewer #1: Dear authors,

This manuscript presents data based on a population cohort to study associations between exposure to antidepressants, their discontinuation, depression and preterm birth, small for gestational and breastfeeding at 6-8 weeks. The data presented are incredibly valuable and valid, however I would recommend to explain and demonstrate why these results are important? How they can contribute to the practice? what can be done regarding depression prevention? The manuscript lacks information, reflection and soundness.

In addition, it seems that authors were just describing results (because data was available) and were not doing it for a specific propose. This feeling begins in introduction. A more reflective way on implications for practice would improve your manuscript.

Please explain better how depression was measured, by whom? What manual/diagnosis system they used? can you guarantee they use the same methods? This need more detail since it is the main outcome.

Reviewer #2: The authors investigated the associations between exposure to antidepressants, their discontinuation, depression and preterm birth, small for gestational age (SGA), breastfeeding at 6-8 weeks. They found that exclusive formula feeding at 6-8 weeks was associated with prescriptions in trimesters 2 or 3 for any antidepressants, SSRIs, particularly higher doses, discontinuation of antidepressants or SSRIs after trimester 1, diagnosis of depression, particularly if medicated, rather than unmedicated. Moreover, preterm birth at <37 and <32 weeks’ gestation was associated with diagnosis of depression, particularly if medicated; birth at <37 weeks was associated with antidepressants, SGA <3rd centile was associated with antidepressants, and SSRIs, particularly higher doses. They stated that both depression and antidepressants indicated increased risks of exclusive formula feeding at 6-8 weeks. These findings will be of interest to practitioners, as well as researchers in the field.

I have the following concerns.

1.Abstract Line 59. Conclusions. What do the authors conclude from the results that intra-uterine exposure to antidepressants and maternal depression were associated with adverse perinatal outcomes? If possible, the authors should comment on the results.

2. The amount of the presented tables is very large, and this is confusing for readers. This aspect of the paper should be improved. Table 2a, 2b,and 2c show the infants excluded from the analysis. If possible, Table 2 could be transferred to the section of supplementary tables.

3. Discussion. Line 275. “To our knowledge, this is the first report that antidepressants, particularly high dose SSRIs, depression, particularly medicated depression, and discontinuation of antidepressants adversely affect breastfeeding rates at 6-8 weeks; the effects of high doses and discontinuation were not independent of depression.”

The authors claim that this is the first report. But what do they mean by “first report”?

Line 423, The authors also stated that “In Wales, the relationships between economic deprivation and breastfeeding, [55, 56] and SSRI prescription in pregnancy, adjusted for diagnosed depression are linear. [48]. This statement indicates that other researchers have also reported the relationships between breastfeeding, [55, 56] and SSRI prescription in pregnancy. The authors should clarify the new findings they found.

In conclusion, I enjoyed reading this paper. This is a valuable paper that presents evidence of the associations between antidepressant prescriptions, discontinuation, depression and perinatal outcomes, including breastfeeding.

6. PLOS authors have the option to publish the peer review history of their article (what does this mean?). If published, this will include your full peer review and any attached files.

Reviewer #1: No

Reviewer #2: No

---

## [Author Response · Author response to Decision Letter 0]

23 Oct 2019

Rebuttal letter PONE-D-19-17104

3.10.19

Dear Professor Hashimoto,

Thank you for the return of our manuscript. We are very grateful for the reviewers’ support and suggestions. We have done our best to comply. 

We hope that readers will concur with reviewer 2: I enjoyed reading this paper. This is a valuable paper that presents evidence of the associations between antidepressant prescriptions, discontinuation, depression and perinatal outcomes, including breastfeeding. 

Thank you for your time spent on this paper. 

Yours,

Sue Jordan, on behalf of all authors

Reviewer #1: 

Dear authors,

This manuscript presents data based on a population cohort to study associations between exposure to antidepressants, their discontinuation, depression and preterm birth, small for gestational and breastfeeding at 6-8 weeks. The data presented are incredibly valuable and valid, however I would recommend to explain and demonstrate why these results are important? Thank you. We have added to lines 455-464:

The concentration of prescribing and adverse outcomes amongst the economically disadvantaged should encourage pre-conception monitoring in tandem with review of antidepressants and non-pharmacological therapies. These data indicate that prescription of antidepressants is an important marker for adverse outcomes, easily identified in primary care records. This could, and should, be used to trigger additional monitoring, including third trimester scans or alternative continuous monitoring technology to detect SGA. Using prescriptions to target care before, during and after pregnancy

How they can contribute to the practice? what can be done regarding depression prevention? We agree, it is essential to explain how the findings contribute to practice. We hope that the implications for practice derived from the study are set out clearly above. Prevention of depression is an important area, but outside the scope of the data presented. 

The manuscript lacks information, reflection and soundness.

In addition, it seems that authors were just describing results (because data was available) and were not doing it for a specific propose. This feeling begins in introduction. 

A more reflective way on implications for practice would improve your manuscript.

 Thank you. To ensure the paper is of a manageable length, readers are referred to earlier work for methodological details at points (references 33 and 17). The information from the study is contained in 9 tables, 5 with 8 multivariate models.

Thank you. We have added to the statement of purpose / aim, line 63 et seq:

… the key question for women and clinicians - the harm to benefit balance of starting, stopping or continuing antidepressant pharmacotherapy [5] - remains unanswered. Accordingly, the aim of this study is to investigate …

The implications for practice are summarised in the ‘Conclusion and Implications’ section, to which we have added, lines 455-464, above. We feel that some caution is needed before drawing causal inferences from observation data, and our main implication for practice is the need to target groups identified for additional care. We have added, line 446:

Drawing causal inferences from observation data may be imprudent. 

Please explain better how depression was measured, by whom? What manual/diagnosis system they used? 

can you guarantee they use the same methods? 

This need more detail since it is the main outcome. Thank you, we have added, line 114-5:

Depression was defined as any diagnosis of depression in the woman’s record before the end of trimester 1 [91 days after 1st day of LMP], recorded by the GP using Read codes, version 2 [17, 36, 37] 

We acknowledge the limitations of using the diagnoses of GPs, lines 598-. 

We acknowledge the risks of under-ascertainment or inconsistency between individual GPs, and the limitations of taking the absence of records as ‘no problems’. 

Thank you. We have added, line 408:

We acknowledge the risks of under-ascertainment or inconsistency between individual GPs, and the limitations of taking the absence of records as ‘no problems’. 

We agree that more detail will improve the value of the paper. Therefore, we have added the S1 file, to give full details of the codes GPs used to record depression. 

Reviewer #2: 

The authors investigated the associations between exposure to antidepressants, their discontinuation, depression and preterm birth, small for gestational age (SGA), breastfeeding at 6-8 weeks. They found that exclusive formula feeding at 6-8 weeks was associated with prescriptions in trimesters 2 or 3 for any antidepressants, SSRIs, particularly higher doses, discontinuation of antidepressants or SSRIs after trimester 1, diagnosis of depression, particularly if medicated, rather than unmedicated. Moreover, preterm birth at <37 and <32 weeks’ gestation was associated with diagnosis of depression, particularly if medicated; birth at <37 weeks was associated with antidepressants, SGA <3rd centile was associated with antidepressants, and SSRIs, particularly higher doses. They stated that both depression and antidepressants indicated increased risks of exclusive formula feeding at 6-8 weeks. Thank you

These findings will be of interest to practitioners, as well as researchers in the field. Thank you

I have the following concerns.

1.Abstract Line 59. Conclusions. What do the authors conclude from the results that intra-uterine exposure to antidepressants and maternal depression were associated with adverse perinatal outcomes? If possible, the authors should comment on the results.

 Thank you. Our comment has been modified, and the section retitled ‘implications’:

Implications: Exposure to antidepressants or depression increased risks of exclusive formula feeding at 6-8 weeks, and prescription of antidepressants was associated with SGA <3rd centile. Prescription of antidepressants offers a useful marker to target additional support and additional care before and during pregnancy and lactation. 

We feel that some caution is needed before drawing causal inferences from observation data, and our main implication for practice is the need to target groups identified for additional care. This is now stated under ‘limitations’. We have added, line 446:

Drawing causal inferences from observation data may be imprudent. 

2. The amount of the presented tables is very large, and this is confusing for readers. This aspect of the paper should be improved. Table 2a, 2b,and 2c show the infants excluded from the analysis. If possible, Table 2 could be transferred to the section of supplementary tables. This is very helpful. Done.

3. Discussion. Line 275. “To our knowledge, this is the first report that antidepressants, particularly high dose SSRIs, depression, particularly medicated depression, and discontinuation of antidepressants adversely affect breastfeeding rates at 6-8 weeks; the effects of high doses and discontinuation were not independent of depression.”

The authors claim that this is the first report. But what do they mean by “first report”?

 Thank you. We have clarified by adding ‘in the published literature’. Line 278:

To our knowledge, this is the first report, in the published literature,

Line 423, The authors also stated that “In Wales, the relationships between economic deprivation and breastfeeding, [55, 56] and SSRI prescription in pregnancy, adjusted for diagnosed depression are linear. [48]. This statement indicates that other researchers have also reported the relationships between breastfeeding, [55, 56] and SSRI prescription in pregnancy. The authors should clarify the new findings they found.

 Thank you, we have clarified, lines 361-3. 

In Wales, linear relationships between economic deprivation and breastfeeding, [58, 59] and economic deprivation and SSRI prescription in pregnancy, adjusted for diagnosed depression, have been identified. [50]

These studies were authored by our group.

In conclusion, I enjoyed reading this paper. This is a valuable paper that presents evidence of the associations between antidepressant prescriptions, discontinuation, depression and perinatal outcomes, including breastfeeding. Thank you. We hope that others will concur, and modify practice to help prevent some of these adverse outcomes in future.

---

## [Decision Letter · Decision Letter 1]

30 Oct 2019

Antidepressant prescriptions, discontinuation, depression and perinatal outcomes, including breastfeeding: a population cohort analysis

PONE-D-19-17104R1

Dear Dr. Jordan,

We are pleased to inform you that your manuscript has been judged scientifically suitable for publication and will be formally accepted for publication once it complies with all outstanding technical requirements.

With kind regards,

Kenji Hashimoto, PhD

Section Editor

PLOS ONE

Additional Editor Comments (optional):

Reviewers' comments:

Reviewer's Responses to Questions

**Comments to the Author**

1. If the authors have adequately addressed your comments raised in a previous round of review and you feel that this manuscript is now acceptable for publication, you may indicate that here to bypass the “Comments to the Author” section, enter your conflict of interest statement in the “Confidential to Editor” section, and submit your "Accept" recommendation.

Reviewer #2: All comments have been addressed

2. Is the manuscript technically sound, and do the data support the conclusions?

Reviewer #2: Yes

3. Has the statistical analysis been performed appropriately and rigorously? 

Reviewer #2: Yes

4. Have the authors made all data underlying the findings in their manuscript fully available?

Reviewer #2: Yes

5. Is the manuscript presented in an intelligible fashion and written in standard English?

Reviewer #2: Yes

6. Review Comments to the Author

Reviewer #2: (No Response)

7. PLOS authors have the option to publish the peer review history of their article (what does this mean?). If published, this will include your full peer review and any attached files.

Reviewer #2: No

---

## [Editor Report · Acceptance letter]

7 Nov 2019

PONE-D-19-17104R1 

Antidepressant prescriptions, discontinuation, depression and perinatal outcomes, including breastfeeding: a population cohort analysis 

Dear Dr. Jordan:

I am pleased to inform you that your manuscript has been deemed suitable for publication in PLOS ONE. Congratulations! Your manuscript is now with our production department. 

With kind regards,

on behalf of

Prof. Kenji Hashimoto 

Section Editor

PLOS ONE